# Raman Enhancement of Nanoparticle Dimers Self-Assembled Using DNA Origami Nanotriangles

**DOI:** 10.3390/molecules26061684

**Published:** 2021-03-17

**Authors:** Sergio Kogikoski, Kosti Tapio, Robert Edler von Zander, Peter Saalfrank, Ilko Bald

**Affiliations:** 1Institute of Chemistry, University of Potsdam, 14476 Potsdam, Germany; kogikoskijunior@uni-potsdam.de (S.K.); kostapio@uni-potsdam.de (K.T.); rozander@uni-potsdam.de (R.E.v.Z.); peter.saalfrank@uni-potsdam.de (P.S.); 2Department of Analytical Chemistry, Institute of Chemistry, State University of Campinas—UNICAMP, P.O. Box 6154, Campinas 13084-974, SP, Brazil

**Keywords:** surface-enhanced Raman scattering, DNA origami, resonance Raman scattering, nanoparticle dimers

## Abstract

Surface-enhanced Raman scattering is a powerful approach to detect molecules at very low concentrations, even up to the single-molecule level. One important aspect of the materials used in such a technique is how much the signal is intensified, quantified by the enhancement factor (*EF*). Herein we obtained the *EFs* for gold nanoparticle dimers of 60 and 80 nm diameter, respectively, self-assembled using DNA origami nanotriangles. Cy5 and TAMRA were used as surface-enhanced Raman scattering (SERS) probes, which enable the observation of individual nanoparticles and dimers. *EF* distributions are determined at four distinct wavelengths based on the measurements of around 1000 individual dimer structures. The obtained results show that the *EFs* for the dimeric assemblies follow a log-normal distribution and are in the range of 10^6^ at 633 nm and that the contribution of the molecular resonance effect to the *EF* is around 2, also showing that the plasmonic resonance is the main source of the observed signal. To support our studies, FDTD simulations of the nanoparticle’s electromagnetic field enhancement has been carried out, as well as calculations of the resonance Raman spectra of the dyes using DFT. We observe a very close agreement between the experimental *EF* distribution and the simulated values.

## 1. Introduction

Surface-enhanced Raman scattering (SERS) is nowadays one of the most used approaches to detect molecules at very low concentrations, even reaching the single-molecule scale [1]. The SERS effect is associated with the presence of very intense confined electromagnetic fields in the junctions of two or more nanoparticles, a so-called hot spot, generated by excitation of the surface plasmon resonance [2,3,4]. Even though it is possible to obtain SERS spectra from single nanoparticles, the minimal structure to obtain a plasmonic hot spot is a nanoparticle dimer, where it is possible to achieve enhancements up to 10^11^ in the nanometric gap between the particles [5,6].

The first report to show that a nanoparticle dimer is a minimal structure to obtain strong Raman enhancements was published in 1999 by Xu et al., using silver nanoparticles modified with hemoglobin (Hb) antibodies, which by properly adjusting the concentration of Hb, self-assembled into dimers containing only one Hb molecule per dimer sitting in the hot spot [7]. Dimeric structures are of fundamental interest for plasmonics, principally because the hot spot of a dimer has a simpler structure, compared to other multi-nanoparticle assemblies and provides a well-defined and intense electromagnetic field [5,8].

The enhancement factor (*EF*) is a critical value used to characterize the SERS performance of a nanostructured substrate [9,10,11,12]. A lot has already been done principally focusing on the necessary *EF* value to achieve single-molecule detection using SERS. Herein we are interested in the distribution of the SERS intensity, and the distribution of *EF*, related to the excitation source. Van Duyne et al. studied the *EF* of dimers using a Raman spectrograph coupled to a transmission electron microscopy (TEM) equipment, showing that the *EF* increases as the wavelength of the laser used in Raman is shifted towards the infra-red region of the spectrum, however only a few dimers were probed due to technically challenging coupled Raman-TEM measurements [13].

DNA origami is nowadays widely used for surface-enhanced spectroscopy studies because it allows the precise orthogonal placement of nanoparticles and other specific anchoring groups, which provide a very straightforward and easy method to obtain plasmonically active ensembles due to the spontaneous self-assembly of DNA [14,15,16,17]. Our group has been working on the development of different DNA-based assemblies for SERS, usually in the dimeric form which was used for single-molecule detection [18,19], or by the creation of nano lenses for dyes or single protein detection [20,21]. More recently a DNA origami nanofork antenna (DONA) was prepared and its application for the SERS detection of single dye and protein molecules was successfully achieved [22].

Herein we will discuss the calculation of enhancement factors for dimers composed of 60 and 80 nm gold nanoparticles (AuNPs) modified with the dyes TAMRA (5-carboxytetramethylrhodamine) and Cy5 (cyanine 5). Both dyes have absorption bands in the regions between 488 and 700 nm, which can give rise to strong SERS signals, both in single nanoparticles and in dimeric assemblies, which enables us to compare the differences in intensity due to the resonance and the hot spot effect. In Figure 1 the experimental and simulated UV-Vis spectra and the calculated molecular structure of both dyes are given. The vertical lines mark the laser wavelengths used, which can couple to an electronic transition giving rise to resonance Raman enhancement. To support our studies finite-difference time-domain (FDTD) simulations of the electromagnetic field enhancement have been carried out, as well as calculations of the resonance Raman spectra of the dyes using density functional theory (DFT). The obtained results show that the contribution of the resonant effect can increase the *EF* compared to a non-resonant condition by 2 times, while the hot spot contributes to further enhance the Raman signal by up to 10^6^ compared to the signal obtained from a single nanoparticle.

## 2. Results and Discussion

### 2.1. Nanoparticle Dimer Self-Assembly Characterization

The schematic of the nanoparticle dimer self-assembly using DNA origami nanotriangles is given in Figure 2A. The nanotriangle is modified in a manner to present sticky ends on both sides of the triangle border so that two DNA-modified nanoparticles can hybridize, one on each side, sandwiching the DNA origami between them. AFM imaging allows the fast and precise observation of the nanostructure shape and size. The AFM images of the DNA nanotriangles showed the correct formation of the nanostructure (data not shown) [23,24]. The AFM images of the self-assembled nanoparticle dimers are shown in Figure 2B for 60 nm dimers and Figure 2C for 80 nm dimers, indicating the appropriate assembly of the dimers while there are still single nanoparticles present over the silicon chip surface.

In Figure 2B, we can observe two well-formed dimers and one single nanoparticle of 60 nm, and in an area of 10 µm^2,^ there are around 10 dimers distributed. For the sample containing 80 nm gold nanoparticles (Figure 2C), the distribution is around 8 dimers in an area of 10 µm^2^. This density allows for the measurement of around 100 individual dimers per analyzed region during Raman mapping. These AFM images show that the dimers are formed, and that the deposition protocol allows the deposition of individual dimers to be studied by Raman mapping.

The samples were also imaged by transmission electron microscopy (TEM) to check the effective gap formation. Some examples of TEM images of the dimers are shown in Figure 2D,E for both the 60 and 80 nm nanoparticle dimers. The TEM shows that the vast majority of the dimers present a gap between the two nanoparticles.

### 2.2. Surface-Enhanced Raman Scattering

The correlation between Raman maps and nanostructures was already obtained previously using SEM, and it proves to be a very straightforward method to study a large number of individual nanostructures [25]. In the work from Van Duyne [13], the correlation between *EF* and excitation wavelengths for 14 dimers were studied and the calculated *EF* varied from 2.5 × 10^4^ at 575 nm to 2.0 × 10^8^ at 785 nm. In the present study, the Raman-SEM correlation was done as shown in Scheme 1, and both the dimers and single nanoparticles which gave enhanced Raman signals are taken into account. First, we compare the average spectra of the dimers and single nanoparticles. In Figure 3 the data obtained for the system containing 60 nm AuNP modified with TAMRA is given. In Figure 4, the data for 60 nm AuNP modified with Cy5 are given. The Raman spectra for the dimers assembled using 80 nm AuNP with TAMRA and Cy5 are presented in the Appendix A (Appendix A).

The SERS spectra for both TAMRA and Cy5 dyes show the expected vibrational bands in the region between 1000 and 1800 cm^−1^, for all the wavelengths used [26,27]. The obtained data show that for the TAMRA molecule the vibrations centered at 1360 and 1650 cm^−1^ are the most intense and are always present using all the excitation lasers, however only the peak at 1360 cm^−1^ was used for the comparison since the peak at 1650 cm^−1^ is located in a region where the performance of the detector is compromised at 785 nm excitation. For Cy5 the chosen peaks are centered around 1380 and 1600 cm^−1^. The intensity shown in the graphs is the peak intensity without the background contribution. The peak intensity shows the same behavior as the integrated peak area.

The initial comparison between the spectra of the single nanoparticles and the dimeric structures shows similar features independent of the chosen dye. For TAMRA and 60 nm nanoparticles (Figure 3), the comparison of counts at 488 nm shows that the dimeric structure only doubles the intensity compared to the single nanoparticle spectrum, indicating the lack of additional enhancement due to the dimer structure in this wavelength. The same is observed at 532 nm, the dimers have double the intensity compared to the single nanoparticle signals, however in this case there is an additional enhancement of the intensity due to the resonance effects of the dye. At 633 nm, it is now possible to observe the effect of the dimeric structure: the dimer exhibits 6.5 times higher intensity compared to the single nanoparticle. However, at this point, it is important to point out that the SERS activity of a single 60 nm AuNP at 633 nm is very low, indicated by the low number of signals coming from single particles. Only 7 single particles showed SERS signals compared to 89 dimers, indicating that at this wavelength plasmonic effects due to the formed hot spots are relevant. At 785 nm, it is not possible to observe any spectra arising from the single nanoparticles, and even for the dimers the number of active dimers is very small, indicated by the very low signal intensity, too, showing that the plasmonic effect is again very weak.

For the system containing Cy5 and 60 nm nanoparticles (Figure 4), the trend is similar again, however, the resonance effect is now coupled to the plasmonic effect at 633 nm excitation, in which the intensity for a dimer is at least 17 times stronger than the single nanoparticle. This multiple is similar to the one observed for 80 nm dimers, indicating that the hotspot is contributing to the enhancement only at one of the investigated wavelengths or only in a certain region of the electromagnetic spectra, showing that such structure cannot be used for a broadband analysis, such as theoretically predicted previously [28].

Since the spectra shown in Figure 3 and Figure 4 are averages, some interesting features of the data are lost. Therefore, histograms containing the intensity of all the measured 60 nm single AuNP and dimers containing Cy5 and TAMRA are plotted in Appendix A, the data obtained from 80 nm AuNP samples are given in Appendix A. From the plot it is possible to observe that such a system has some particularities for the dimers, i.e., we observe a log-normal distribution of intensities (in Appendix A the curves appear to follow a normal distribution due to the logarithmic scale on the intensity axis). A log-normal distribution indicates a non-symmetrical distribution of intensity. This inhomogeneity can come from several factors, such as the gap distance variation, the nanoparticle size distribution, the crystal orientation of the nanoparticles, or other plasmonic effects such as plasmon-induced reactions [29,30,31]. The distribution of intensities in SERS experiments is supposed to follow a long-tail distribution, i.e., most of the signals (80%) are centered around a particular intensity range, while the other 20% occur with very high intensities [10,32]. The log-normal distribution can be considered a long-tail distribution curve. From our analysis, we conclude that the dimers also present a long-tail intensity distribution.

It is possible to observe that at 633 nm the signal intensity is much stronger compared to the other wavelengths, a very good indication that for the case of a dimer, the hot spot is only active at one of the considered excitation frequencies. Of course, with more excitation lines available it would be possible to achieve a spectral distribution of enhanced intensities more precisely, such distribution so far has only been reported through simulations and very few experimental attempts [28,33,34].

The distribution of intensities follows the same trend as described before: For dimers containing TAMRA (Appendix A), a very low-intensity signal is observed at 488 nm. At 532 nm, TAMRA is resonant with the excitation laser resulting in resonance enhancement, and at 633 nm the plasmon electromagnetic enhancement yields very intense SERS signals, 3 to 10 times stronger compared to 532 nm. At 785 nm the signal is again very weak. For Cy5 the signals at 633 nm are 30 to 750 times stronger than at 532 nm due to the simultaneous resonance enhancement provided by the molecule and the plasmonic enhancement. This shows that the proper choice of the excitation laser, probe molecule, and dimer structure can lead to a much higher intensity.

To explain the plasmonic enhancement, we used FDTD simulations to visualize the enhanced electromagnetic field in the dimers, and also to obtain the distribution of electromagnetic enhancement for the dimers in the spectral range studied here. The electric field enhancement map using 633 nm as the excitation source and the *EF* distribution in the visible spectrum for a 60 nm dimer is given in Figure 5. The maps for the other excitations are given in the SI. In the simulations, a gap distance of 3.5 nm is assumed, which is based on the DNA double helix diameter of ~2 nm [35,36], and a DNA shell thickness of in total 1.5 nm to account for the nanoparticle surface modification. From the maps, we can observe that the maximum field enhancement is located near the particle surface and that the maximum is near the axis of the dimer. From these maps, we can obtain a theoretical hot spot area which will be important later on.

The simulated field enhancements for all the wavelengths used for the Raman analysis are given in Appendix A of the Appendix A. Considering the hot spot to be a rectangle and a minimum |E/E_0_| value of 120 to delimit the maximum area, we obtain for the 60 nm dimers ~24 nm^2^ and the 80 nm dimer ~56 nm^2^ of the hot spot area at 633 nm (Appendix A).

The simulated *EF* showed in Figure 5C,D helps us to understand the signal intensity observed experimentally for both 60 and 80 nm single nanoparticles and dimers. For single particles, the maximum possible *EF* is obtained at 532 and 560 nm for 60 and 80 nm nanoparticles, respectively (*EF* showed in the graph). For the other wavelengths studied here, the *EF* of single nanoparticles is well below 10^3^. This low *EF* is the reason why we observe few SERS active single particles using 633 nm and TAMRA, and we cannot observe any signals at 488 nm for Cy5, and at 785 nm for all the cases (absence of resonance from the dyes and weak plasmonic activity for the nanoparticles).

On the other hand, for the dimeric nanostructures, we observe a maximum *EF* around 10^6^. At 488 nm the simulated *EF* is in the order of 10^2^, which is very similar to the *EF* of single nanoparticles. At 532 nm the *EF* of the dimer is around 5 × 10^3^. Therefore, this indicates that the dimer structure is not contributing significantly to increase the signal at those two wavelengths. Since the *EF* of the single nanoparticle is approximately the same as the *EF* of the dimer, we can only observe a doubling of the signal intensity due to the dimeric structure (two particles = twice the amount of dyes). At 633 nm the calculated *EF* is 10^6^ for the 60 nm AuNP dimer and 10^5^ for the 80 nm, which is near the maximum of the simulated curves, for 80 nm the maximum *EF* is around 700 nm. For 785 nm the *EF* decreases for both 60 and 80 nm dimers to 10^4^ and 10^5^, respectively. At 785 nm, we cannot measure single particles showing the very low plasmonic activity of them at this wavelength, but we could observe some dimers, an indication that at this wavelength the hot spot can still be active.

For analytical purposes, the enhancement factor can be defined by normalizing the intensity of the Raman bands with and without nanoparticles [9,37]. However, for our comparison we can obtain the amount of dye on the nanoparticle surface, and instead of comparing the dimer signal to the normal Raman intensities, it is more convenient to compare the dimer signal to the signal obtained from single particles modified with the dyes. The *EF* then can be defined by Equation (1) [20]:(1)$$EF_{dimer}=\left(\frac{I_{dimer}}{I_{single}}\right)\times \left(\frac{N_{single}}{N_{dimer}}\right)\times EF_{single}$$

In this equation, the ratio between the dimer and single-particle intensity (*I*) is multiplied by the ratio of the number of molecules in the single nanoparticle and the dimer (*N*) multiplied by the *EF* of a single nanoparticle. All those variables can be measured and the *EF* for a single particle can be estimated. In our case measuring the Raman spectra of the dyes without nanoparticles present turned out to be a challenge due to a very high fluorescence, which is quenched near the metallic nanoparticle [38].

At this point, it is important to present some considerations regarding the *N_dimer_* and *EF_single_*. Considering the experimental results obtained we observe that for 488 and 532 nm the intensity of the dimer is double as high as that of the single-particle, indicating that in those situations all the molecules present at the surface of the nanoparticle are contributing to the obtained signal, so that *N*_dimer_ = 2 × *N*_single_. However, at 633 nm, we observe that the dimers have a much stronger intensity and that the molecules in the hot spot of the nanostructure are contributing more strongly to the signal generation. Considering that the hot spot in such situation is confined to a small fraction of the nanoparticles and only those molecules are generating the signal, and considering that the number of molecules in the hot spot is the dye surface density multiplied by the hot spot area, $N_{dimer}=2\times \left(\frac{N_{single}}{SA_{single}}\right)\times HS_{area}$, where *SA_single_* is the surface area of a nanoparticle, and *HS_area_* is the area of the hot spot in the dimer. Substituting this in Equation (1) we can obtain Equation (2) which can be rearrange to generate the final form Equation (3):(2)$$EF_{dimer}=\left(\frac{I_{dimer}}{I_{single}}\right)\;\times \;\left(\frac{N_{single}}{2*\left(\frac{N_{single}}{SA_{single}}\right)*HS_{area}}\right)\;\times \;EF_{single},$$
(3)$$EF_{dimer}=\left(\frac{I_{dimer}}{I_{single}}\right)\;\times \;\left(\frac{SA_{single}}{2*HS_{area}}\right)\;\times \;EF_{single},$$
we can observe that for a dimeric structure with nanoparticle fully covered by a monolayer of dyes the enhancement effect is not dependent on the amount of dye on the surface. Considering that there is the formation of a plasmonic hot spot, the most important parameter for the enhancement is the area of the hot spot, such that, a very small hot spot would increase the *EF* substantially, indicating that smaller gaps in a dimer produce greater enhancements. The *SA_single_* was calculated to be 11310 nm^2^ for the 60 nm nanoparticle and 20100 nm^2^ for the 80 nm nanoparticle. The *HS_area_* used for the calculation was given above to be 24 nm^2^ for 60 nm dimers and 56 nm^2^ for the 80 nm dimers.

Obtaining the value of *EF_single_* using experimental approaches is challenging because it requires the measurement of the dyes in solution at each excitation wavelength used, but we could not obtain the spectra of the used dyes due to fluorescence of the molecules which suppresses any Raman scattering. In the literature, the values obtained for the *EF* of single gold nanoparticles with 60 and 80 nm range between 1 × 10^5^ and 1 × 10^6^ [39]. This value is also wavelength-dependent, but there is no data comparing the SERS efficacy of single nanoparticles at different wavelengths. Since these values are usually obtained at only one wavelength, and under experimental conditions that are not similar to the ones used by us, we decided to use the *EF* obtained from the simulations presented in Figure 5C.

Using the consideration given above in Equation (3) and the average intensities of the single nanoparticles obtained from the plots of Figure 3 and Figure 4 we summarize the *EF* distributions in Figure 6. From the data we can observe that only the 633 nm laser is very efficient in generating a strong SERS effect due to the plasmonic hot spot, reaching an average enhancement in the order of 10^6^ for both 60 and 80 nm nanoparticles with all the dyes. For the dimers containing TAMRA, there is an additional resonance enhancement at 488 and 532 nm. At these wavelengths the molecule’s fluorescence is quenched by the metal nanoparticle, however, the obtained values are still in the range of *EF* of the single nanoparticle (comparison with the simulation date from Figure 5C). For Cy5 there is no resonance effect taking part at 488 nm, and no signal was observed, and at 532 nm the enhancement is the same as that of a single AuNP. At 633 nm, for Cy5 the obtained value of *EF* is approximately double as high as that of the system containing TAMRA. This indicates that the molecular resonance increases the signal by a factor of 2, similar to the factor observed for TAMRA under resonant conditions (488 and 532 nm).

For the 60 nm TAMRA case at 633 nm, the *EF* might be underestimated, because there is one large uncertainty associated with a low number of single particles measured. This is due to the low plasmonic activity of the single particles, and consequently the detected intensity values *I_single_* represent the highest intensities within the whole distribution and thus can be overestimated, which decreases the *EF* proportionally. Furthermore, some values were not possible to be determined due to a lack of a SERS spectrum for the case of Cy5 at 488 nm, or for all cases when using 785 nm, since we could not obtain any spectra coming from single nanoparticles possibly due to a low *EF_single_*. For this second case, we see that the number of dimers generating SERS spectra is also very small, an indication that the *EF*, in this case, should be small, and only the dimers at the edge of the statistical distribution are observed.

### 2.3. Resonance Raman Spectra Calculation

The resonance Raman enhancement is a fundamental part of this study, however as observed in the UV-Vis spectra of Figure 1, the regions where resonance enhancement could be observed are limited. Moreover, it is very challenging to obtain the Raman spectra of the studied dyes at the wavelengths we used, principally due to strong fluorescence emission from the molecules, in this way it is uncertain of the effect of resonance enhancement on the appearance of the SERS spectra. Although in SERS the fluorescence is typically quenched by the metal nanoparticles, the situation is further complicated by the fact that in SERS sometimes only some vibrations and not the full spectrum are enhanced [40,41].

Here we calculated resonance Raman spectra of TAMRA and Cy5 at different excitation energies, to check spectral variations depending on the excitation energy. Normal Raman and resonance Raman (rR) spectra were calculated from DFT, taking water as a solvent into account via a continuum model. Differences in rR spectra as a function of excitation wavelength can help in the analysis of the *EF* since not all peaks present the same enhancement. In Figure 7, a comparison between the experimentally obtained SERS spectra at 633 nm, and the calculated rR spectra for both TAMRA and Cy5 at selected energies are shown. Here we note that, according to time-dependent DFT calculations, the vertical excitation energies into the lowest bright singlet states of TAMRA and Cy5 are at 462 nm (2.69 eV) and 529 nm (2.34 eV), respectively (see Figure 1 and the section on computational methodology).

The results presented in Figure 7A,B show a very good correlation between the simulated rR spectra and experimental SERS spectra when both TAMRA and Cy5 were excited by the 633 nm laser in the experiment. In contrast, the experimental spectra are quite different from the simulated normal (i.e., non-resonant) Raman spectra. This indicates that indeed we are performing surface-enhanced resonant Raman scattering (SERRS) even when out of the nominal resonant conditions of the molecules. This is particularly true for TAMRA, where the excitation wavelength of the free dye is around 550 nm according to Figure 1. We generally observe a very good agreement between the most intense peaks, with a difference in peak center position of about 20 to 30 cm^−1^, which is attributed to the simplified computational model and method.

Since for 633 nm, where free TAMRA is considered out of resonance, the SERS spectrum still is in good agreement with the simulated resonant spectrum, the presence of the metal nanoparticles red-shifts the absorption enabling the resonance condition.

From the simulated data shown in Figure 7C,D, we observe that the resonance Raman spectra have high intensities only in a region of the visible spectrum close to (and somewhat below) the vertical excitation energies to the first bright singlet states. For TAMRA, the resonance effect is observed between 516 and 460 nm, and for Cy5 the resonance is most important between 652 and 539 nm. While in good agreement with computed vertical excitation energies, those wavelengths do not match exactly the experimental absorptions but are close enough to directly correlate to our experimental SERS data. Note also that the computations were done for free molecules in a water solvent, without the presence of nearby nanoparticles, DNA, or ions. Although it is well-known that the molecular orientation with respect to the surface plays an important role for SERS, the dye molecules in the present experiments are not directly adsorbed to the nanoparticles, but bound to the DNA via linkers. Therefore, they are relatively free to rotate. Furthermore, in the experiments many molecules are detected within one measurement, i.e., the spectra represent an average conformation of the molecules within the nanoparticle gap. Consequently, the experimental spectra are comparable with the calculated spectra, which do not take the surface directly into the account and which are averages over all molecular orientations [4,41]. Another point is that we do not observe substantial peak shifts due to the different excitation wavelengths in the simulations, in accordance with our SERS data, where the peak positions are largely independent of the laser used.

Another important aspect that can be observed is that different vibrations are enhanced in different regions of the spectra. For example, for Cy5, in Figure 7D, the rR peak at 1142 cm^−1^ is the strongest in the region between 620 and 577 nm, while at 563 nm, i.e., closer to the resonance, other peaks present in the spectra are now very intense, e.g., the peak at 1380 cm^−1^. In the normal Raman spectrum, the most intense peak is the one at 1634 cm^−1^. Similar behavior is observed for TAMRA but in a more subtle way, because the rR peak intensities fluctuate depending on the excitation energy. This is a difficulty since many SERS studies rely on dyes, making the comparison between different SERS substrates difficult. We should note that for TAMRA the stronger correlation between SERS spectra at 633 nm and computed rR spectra in the resonance region compared to normal Raman spectra, is slightly less striking than for Cy5 but still visible. This could be because at 633 nm, we are farther away from resonance in the case of TAMRA than for Cy5.

Other features we can see are related to the intensity of the vibrations, those at low-frequency region (below 1000 cm^−1^) are more intense compared to those at higher wavenumbers, however, in our experiments the vibrations at low frequencies are not observed, indicating that the dimers better enhance the vibrations at higher frequencies.

## 3. Conclusions

The use of DNA origami allowed us to easily create plasmonic devices arranging nanoparticles very precisely. Compared to other methods to produce plasmonic devices, DNA origami is cheaper, faster, and scalable with very good control over nanoparticle placement. The use of Raman and SEM correlation proved to be fruitful to analyze contributions from both single nanoparticles and self-assembled dimers. The SERS results showed that the distribution of intensities of around 1000 individually measured dimers follows a log-normal distribution, with some dimers presenting a very strong enhancement, probably due to smaller gaps between particles or other possible defects from the nanoparticle surface.

From the SERS data, we can also conclude that plasmonic effects of the dimers occur mainly at one wavelength, 633 nm, while at the other wavelengths only the plasmonic enhancement of the single nanoparticles contributes as well as the molecular resonance enhancement. From the *EF*s distribution, we observe that the resonance effects in our case contributed with a factor of 2, i.e., the *EF* in SERRS is doubled compared to the *EF*s of the system out of resonance, similar to values reported in the literature. When the excitation light is in resonance with the dimer and the molecular absorption, enhancements in the order of 10^6^ are observed, but it is important to emphasize that this value is dependent on the probe molecule, excitation wavelength used, and the nanoparticle assembly.

The resonance Raman simulations helped in the characterization of the system, the simulation of both Cy5 and TAMRA dyes resulted in spectra in good agreement with the experimentally obtained ones. We also found that the experimental SERS spectra recorded under conditions out of resonance of the free dye molecules still closely resemble the calculated resonance Raman spectra, an indication that possible charge transfers between the nanoparticles and the dyes are occurring, shifting the resonance towards the red region of the spectrum.

## 4. Experimental Section and Computational Details

### 4.1. Gold Nanoparticles Modification Procedure

The modification of gold nanoparticles was realized using procedures from the literature [42,43]. Initially 1 mL of 60 or 80 nm AuNP suspension was concentrated to 100 µL by centrifugation, this step also removed excessive stabilizing agents present in the solution. To that solution, 25 µL of 2.5 mmol of Bis-(p-sulfonatophenyl)phenylphosphine (BSPP) was added and shaken for 1 h at 37 °C. After that sodium dodecyl sulfate (SDS), was added at a final concentration of 0.02%. Then the thiolated DNA sequences were added in excess, depending on nanoparticle size, for 60 nm 6000× excess and for 80 nm 8000× excess, again the solution was shaken for 1 h to allow the DNA-SH to interact with the nanoparticle surface. To this solution, NaCl 2 mol L^−1^ was slowly added to the final concentration of 0.7 mol L^−1^ (10 to 20 additions) in the course of 6 h, at the final concentration the solution was left shaking overnight. On the other day, 150 µL of 1× TAE buffer with 15 mmol MgCl_2_ was added to the nanoparticle suspension, and interacted for 1 h, to prevent aggregation of the nanoparticles while washing the suspension and later when incubated with DNA origami. The modified nanoparticle suspension was washed with 1× TAE buffer with 15 mmol L^−1^ MgCl_2_ and 0.02% SDS, 5× by centrifugation, if aggregation was observed during the washing step the nanoparticles was sonicated for 1 min at 37 °C, this washing step removed all excess DNA which could hinder the dimer self-assembly. This modified nanoparticle suspension was used for the dimer assembly right after the washing step.

### 4.2. DNA Origami Synthesis and Dimer Formation

Triangular DNA origami was formed according to a modified version of Rothemund’s method [19,20], the M13mp18 viral DNA (5 nmol L^−1^) is the scaffold sequence, and the modified staples were mixed in a 1:30 ratio in a solution containing 1× TAE buffer, with 15 mmol L^−1^ MgCl_2_ to a total volume of 100 µL. The triangle structure contains 215 different DNA staple sequences, and from those, we chose 8 sequences to be modified to contain the complementary sequence to the one presented at the modified nanoparticles, at positions that the nanoparticles would have minimum separation distance.

The annealing was performed in a thermocycler first by increasing the temperature of the solution to 90 °C, and gradually decreasing the temperature from 90 to 15 °C over the course of 2 h. After the annealing, the DNA origami was purified using an Amicon Ultra-0.5 mL filter (100,000 Da MWCO, 300× *g* speed, 10 min) followed by washing with 1× TAE-MgCl_2_ (300 µL) solution to get rid of the excess staple strands, the washing step was repeated 5 times. All DNA oligos were purchased from Metabion GmbH (Martinsried, Germany). All oligos were either HPLC purified or FCP purified (unmodified staple strands).

To the purified DNA origami, the modified gold nanoparticles were added in a 3:1 ratio (nanoparticle: Origami) and incubated for 2 h at 45 °C and cooling to 20 °C for 2 h (total time 4 h). This step enables the formation of the self-assembled dimer, but it forms a mixture of other aggregates that are not desired for the SERS analysis. Then the sample is purified by gel electrophoresis, using a 1% (*w/v*) of agarose in 1× TAE buffer with 15 mmol L^−1^ of MgCl_2_. The gel is cut, and the fraction containing dimers is separated. From these portions, the assemblies are deposited over plasma-treated SiO_2_ chips.

From the solution of purified dimers, 2 µL is dropped cast over the substrate, and 15 µL of 5× TAE with 75 mmol L^-1^ MgCl_2_ is added to help the adsorption of the dimers on the surface. The solution is left interacting with the SiO_2_ for 1 h at a humid chamber. Then the substrate is washed with 1:1 (*v:v*) H_2_O: Ethanol solution three times, and blown dry with N_2_. After this, the materials are ready to further use.

### 4.3. SERS Mapping, SEM, TEM and AFM Characterization, and Correlation Procedure

SERS spectra have been recorded using a confocal Raman microscope (WITec 300α) equipped with an upright optical microscope. For Raman excitation, laser light at λ = 488, 532, 633, and 785 nm was used, which were coupled into single-mode optical fiber and focused through a 100× objective (Olympus MPlanFL N, NA = 0.9) to a diffraction-limited spot between 650 to 1000 nm^2^. The laser power was set to 300 µW at the focal plane, and the integration time was 2 s. The deposition procedure explained before was chosen such that about 10 dimer assemblies were located at a 10 × 10 µm area. The images varied between 40 × 40 to 80 × 80 µm, to measure around 50 to 100 different nanostructures. Care was taken to avoid photodamage of the samples.

After the Raman mapping, scanning electron microscopy (SEM) images were obtained using the equipment Thermo Fisher Phenon ProX Desktop SEM or a FEI Quanta 250. Transmission electron microscopy (TEM) images were obtained using the equipment JEOL JEM 1011, equipped with an Olympus MegaView G2 camera, and using 80 kV acceleration voltage. The dimers were deposited over copper grids containing a 1 nm carbon layer on top of 10 nm formvar film (EFCF400-Cu-50, Science Services GmbH), on the grids 3 µL of the dimer stock solution was deposited and incubated for 2–3 min. The excess solution was removed, and the grid was washed twice with Milli-Q water. After drying at room temperature, the grid is ready for imaging. Atomic force microscopy (AFM) images were obtained using a Bruker MultiMode 8 and a Nanosurf FlexAFM in the air using tapping mode.

The correlation between the Raman maps and the SEM images is done in some steps. First, the dimer is deposited over a SiO_2_ marked with a cross, and then the Raman maps are obtained in a region where it is possible to observe features of the SiO_2_ that can be imaged by SEM, serving as a dual guide. At this point, the Raman data is obtained, and later the substrate is imaged using SEM with high resolution, then the SEM images and the Raman mapping are assembled to correlate the SiO_2_ vibrations with the scratches present in the substrate. With the positioning done, the Raman map is substituted for the SERS signal map, and the correlation between the dimer observed by SEM and the SERS signal is obtained. This process is depicted in Scheme 1. From the correlation data, isolated signals arising from single nanoparticles and dimers are collected for further analysis. All the correlations and extracted data are given in the Appendix A.

### 4.4. FTDT Simulations

In brief, we used the Lumerical Finite Difference Time Domain (FDTD) Solutions software (v8.19.1584), provided by Lumerical Inc., Vancouver, Canada. The model included two gold spheres (refractive index from Johnson and Christy) [44] surrounded by a 1.12 nm DNA layer (refractive index 1.7) [45] and on top of a silicon substrate (refractive index from ref. 45) [46] with 3 nm silicon dioxide (refractive index 1.44) on top of the Si. The DNA layer thickness was measured using AFM by Tapio et al. [22]. Here, between the particles, a set of double-strand DNA (refractive index 2.1) [45] is added to simulate the triangular DNA origami. The medium was defined as air. The distance between the surfaces of the gold spheres was set at 3.5 nm. The E_field_ distributions were calculated and the field maximum around the gap region was recorded. For the average E_field_ distribution, we recorded E_field_ distribution around the particle surfaces and only considered cases, where the field enhancement was higher than 5 |E/E_0_|. The area that qualified this criterion can be covered by a cone starting from the middle of the particle and which has roughly the opening angle of 58°. To solve the E_0_ case, we removed the DNA layers and the metal spheres. The incident of light was perpendicular to the substrate and the electric field polarization along the gap (the gap mode) was considered.

### 4.5. Theoretical Methods for (Resonance) Raman Simulations

Resonance Raman and ordinary Raman spectra were calculated by hybrid Density Functional Theory (DFT). Specifically, the B3LYP exchange-correlation functional [47,48] and the 6-311++G** basis set [49,50,51,52] were used in connection with the Grimme D3 dispersion correction. [53] Water as a solvent was modeled by the polarizable Continuum Model (PCM) [54,55].

In the first step, geometry optimizations without any symmetry restrictions were performed for TAMRA and Cy5 using the Gaussian 16 program package [56]. Slightly simplified molecular models compared to those in the experiment were used and are shown in Figure 1. Specifically, the Cy5 complex -SO_3_H groups were replaced by H and the CH_2_-(C_3_H_6_)-DNA connecting unit by -CH_3_. For TAMRA, the -C(O)-DNA connecting unit was replaced by -C(O)-NH-CH_3_.

In a second step, ground-state frequency analyses were performed for both molecules, and corresponding ordinary Raman spectra were calculated. No scaling of frequencies was done. For TAMRA, we have 168 normal modes and 174 for Cy5. The stick spectra were broadened by Lorentzians with a Full Width at Half Maximum (FWHM) of 8 cm^−1^.

In a third step, electronically excited singlet states (excitation energies and oscillator strengths) were calculated, for both molecules, by the linear-response time-dependent DFT method (TD-B3LYP+D3/6-311++G**+PCM (water)), also using Gaussian 16. The lowest, bright excited singlet states are found at 2.69 eV (462 nm) for TAMRA (with an oscillator strength of 0.887), and at 2.34 eV (529 nm) for Cy5 (with an oscillator strength of 2.097). Computed, broadened absorption spectra are shown in Figure 1B. Broadening with Lorentzians with FWHM = 20 nm was adopted in this case.

In a final step, resonance Raman spectra were calculated using the Independent-Mode-Displaced-Harmonic-Oscillator model in Short-Time-Approximation (IMDHO-STA), [57] all in harmonic approximation for the vibrations, using a home-made program. [58] This method is described in detail in Ref. 57 and references cited therein. It makes use of a time-dependent cross-correlation function approach to calculate the Raman polarizability for vibrational transitions between initial and final vibrational states in the electronic ground state, via an intermediate electronically excited state, which we take to be the lowest bright singlet state for each molecule calculated by TD-DFT. In the STA, no extra excited-state optimization including normal-mode analysis is performed, instead one uses the same vibrational frequencies and normal modes as in the ground state, together with excited-state displacements and dimensionless shifts for each mode calculated from the excited state gradients along the normal modes. Thus, the correlation function is evaluated from ground state properties and excited-state gradients only, this way neglecting frequency alterations in the excited state and/or Duschinsky rotations but including the shift of the excited state. The polarizabilities are then obtained from a Fourier transformation of the (damped) cross correlation function, where the Fourier transform also accounts for the vertical electronic excitation energy. Finally, the rR intensities (the “Raman shifts”), are calculated from:(4)$$\sigma_{rR}^{i\to f}\left(\omega_{L},\omega_{S}\right)=\frac{8\pi \omega_{L}\omega_{S}^{3}}{9c^{4}}\underset{q,q^{\prime }}{\sum }\left|{\left.\alpha_{i\to f}^{qq^{\prime }}\left(\omega_{L}\right)\right|}^{2}\right.$$
where *i* = 0 and *f* = 1 denote ground state vibrational quantum numbers for the fundamental Raman transitions, *α* is the Raman polarizability tensor (with component indices *q*,*q*’ = x,y,z), and *c* the velocity of light. *ω_L_* and *ω_S_* are the frequency of the exciting light and the scattered radiation, respectively. The exciting light frequency was systematically varied in the different spectra shown in Figure 7. One obtains a stick spectrum with one stick corresponding to each mode. In practice, two Lorentzian broadening factors are involved: One for the damping of the correlation function, for which we adopt a FWHM of 150 cm^−1^ in this work, and one for the rR stick spectra, for which we adopt FWHM = 8 cm^−1^ for Cy5 and 16 cm^−1^ for TAMRA, respectively.

### 4.6. Data Analysis

The obtained results were analyzed using the software made available by the equipment manufacturer, such as Project 5.0 used for the analysis and plot of Raman maps, and collection of the spectra from Witec. The plots, graphical and statistical analysis were all obtained using Origin^®^ 2020 tools. AFM images were analyzed using Gwyddion^®^ and TEM using ImageJ^®^, which are open source and freely available software.

## Data Availability

The data reported here is available from the authors upon request.

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
