# Peer review of "Raman Enhancement of Nanoparticle Dimers Self-Assembled Using DNA Origami Nanotriangles"

_molecules, 2021, doi:10.3390/molecules26061684_

Round 1

Reviewer 1 Report

this manuscript reports on Raman enhancement calculation on nanoparticle and nanoparticle dimers prepared by using DNA origami.

the topic is rather interesting and several groups are still working on this approach for SERS and enhanced spectroscopies in general. 

The authors discussed here experimental and theoretical results on two dyes Cy5 and TAMRA used as SERS probes and they looked at the emission from individual particles or dimers. 

The manuscript is written rather well, even if several errors can be find along the text. in particular all the exponential number (10^6, 10^5 etc) are wrongly reported (appear as 106, 105, 108 etc)

from the technical point of view, on the contrary, the manuscript is quite obvious. the authors claim that their "results show that the calculation of the EF is not only a concentration dependent relationship, but it is a measurement dependent on the resonance effect, the nanoparticle coupling, and principally on the size of the hot-spot obtained" This sentence is not absolutely a new results. every person that works in the Raman field know that the EF in SERS depends on different parameters, so it is really not clear to me where is the novelty here.

in figure 4 the authors discussed the ehnacement due to single particle and dimers. they say that the intensity are double etc, but this is a big semplification. the spectra must be analyzed considering not the peak intensity, but the area below the peak also normalizing with respect to the different background. Moreover, it's again absolutely obvious that with a dimer you can have an enhancement in the SERS signal.

Figure 5 wants to discuss some simulated data, anyway, the quality of the simulation is terribly bad. panels A and B should report the e.m. map but it's clear that the used mesh is not small enough. the authors also say, few lines below the figure "the EF is much smaller below 103, which do not really increase the Raman signal" . i suppose that it's 10^3...but I cannot understand why an enhancement of 1.000 should not increase the raman signal!!!!!

there are also 2 equations reported in the text, but the font used is really too large and the alignement of the text should be corrected.

at page 9 the authors say "even though it is more difficult to insert a molecule into very small gaps...." this is absolutely not true! there are tons of examples on that (see the recommend refs below)

figure 6 reports EF distribution. to me the use of lines to connect the different points is not correct and should be removed.

in section 2.3 the authors discuss about resonance raman spectra calculation. this is probably the only really interesting point of the manuscript, even if also this method is really well known. anyway the analysis is limited and the authors should mention additional issues. first of all I don't see the rationale to use here energy in eV while in the manuscript everything is discussed in nm. moreover, much more importantly, the authors are missing a fundamental point in SERS measurements and calculation. the observed peak intensities can be strongly dependent from the orientation of the molecule on the metallic surface. strongly localized field, in particular in narrow nanogap can ehnance sub-section of the molecule and consequently some vibration more than others. There is an extensive very recent literature on that (see for example Angew. Chem., Int. Ed., 2020, 59(28), 11423–11431 and other mentioned below)

in the conclusion the authors say "From the SERS data, we can also conclude that plasmonic effects of the dimers occur mainly at one wavelength, 633 nm, while at the other wavelengths there is only a doubled signal, showing that the incident radiation is important for the generation of the so-called hot-spot." which is the meaning of "double signal"??? again it's absolutely obvious that the incident radiation generate an hot-spot...and that the hot-spot plays the major role in the ehnancement for SERS.

in conclusion I think that in order to be published, also as a technical paper, the manuscript need to be significantly improved, mainly considering the existing literature that seems to miss here. the authors must revise the paper underlying the novel points they want to discuss and avoiding basic and well-known concepts. 

recommended literature to be included in the discussion:

nanocavities for SERS

-Nat. Mater., 2019, 18(7), 668–678

-Nanoscale Adv., 2021, 3, 633

-Nat. Commun., 2019, 10, 5321

-ACS Nano, 2016, 10(11), 9809–9815

-Science, 2016,-354, 726–729.

DNA origami enhanced spectroscopies

-ACS Photonics, 2018, 5(4), 1151–1163.

-Adv. Funct. Mater., 2018, 28(15), 1707309.

-Nano Lett., 2019, 19(9), 6629–6634.

-ACS Nano, 2018, 12(2), 1650–1655

-J. Am. Chem. Soc., 2017, 139(48), 17639– 17648.

-Chem., Int. Ed., 2018, 57(11), 2846–2850.

-Nano Lett., 2018, 18(1),
405–411.

Reviewer 2 Report

The paper is interesting, full of interning and useful information and will be well cited in the future.  Authors presented clear evidence how several characteristics and conditions could have influence on the intensity and appearance of the Raman peaks (Raman Enhancement effect). Although the things related to this subject were investigated and reported even before and although the response on questions asked at the principio of the paper are not given in complete , the paper offers  many interesting information and will be very useful for further  discussions and development in the field.   It is the high quality paper in which all studies are designed and performed at very precise and systematic way. Authors used several   wavelengths: 488 nm, 532 nm, 633 nm and 785 nm, the same laser intensity (important), obtained AuNPs with well-defined surface properties and characterized with many different techniques.  The experimental findings and theoretical models (calculations) are in good agreement and the whole paper is presented around defined and single   idea. Because of this it is easy to follow, has lot of sense and potential on the high impact level.  It should be publish as it is. 

Author Response

We thank the reviewer for the positive assessment of our manuscript.

Reviewer 3 Report

In this manuscript, Kogikoski Jr. et al. exploits the nano-triangular DNA origami to assemble gold nanoparticle dimers for surface enhanced Raman scattering (SERS). It is reported that a maximal enhancement appears at 633 nm as the plasmonic resonance of the obtained dimers occurs at this wavelength. Moreover, the authors further demonstrate that the enhancement factor (EF) is not only dependent on the excitation wavelength but the probe molecules as well as the nanoparticle assembly. Nevertheless, there are still some points needed to be considered:

  1. There are quite some typos in the main text. For example, contribu-tion, ob-served, which are found in Abstract. Please double check the manuscript and make appropriate modification.
  2. As pointed out in this manuscript, the size of the hot spot can affect the EF. Is it possible to tune the gap size by adjusting the thickness of DNA shell to see how it works at different gap size?
  3. To further verify the role of molecular resonance effect in SERS, it will be good to select some other molecules with their resonant peak at different wavelengths, e.g., 488 nm or 532 nm, and check the SERS performance.

Round 2

Reviewer 1 Report

The authors replied to all my concerns and improved the manuscript accordingly

I recommend the publication in Molecules

Reviewer 3 Report

All my questions have been addressed. I recommend this paper for publication.